# Impact of Ultraviolet Radiation on Growth, Development and Antioxidant Enzymes of *Tuta absoluta* (Meyrick)

**DOI:** 10.3390/insects16020109

**Published:** 2025-01-22

**Authors:** Junhui Zhou, Wenfang Luo, Wei He, Xin Huang, Suqin Song, Liang Mao, Huan Peng, Jianjun Xu

**Affiliations:** 1Laboratory of Integrated Pest Management on Crops in Northwestern Oasis, Ministry of Agriculture and Rural Affairs, Xinjiang Key Laboratory of Agricultural Biosafety, Institute of Plant Protection, Xinjiang Academy of Agricultural Sciences, Urumqi 830091, China; junhuiqzhou@163.com (J.Z.); lf576263465@163.com (W.L.); hewei8299@163.com (W.H.); huangxin0924@126.com (X.H.); suqin_song@163.com (S.S.); 2Tulufan Agricultural Technology Extension Center, Tulufan 838000, China; mao2548@163.com

**Keywords:** antioxidant activity, longevity, reproduction, *Tuta absoluta*, ultraviolet stress

## Abstract

*Tuta absoluta* Meyrick (Lepidoptera: Gelechiidae) is an invasive and destructive pest, posing a significant threat to solanaceous crops. Ultraviolet radiation (UV) is a prominent abiotic stressor for numerous organisms, including insects. However, limited evidence suggests that UV radiation (including UV-A, UV-B, and UV-C) plays a significant role in affecting *T. absoluta*. It was hypothesized that ultraviolet radiation could impact *T. absoluta*. In this study, the results suggested that UV-A and UV-C radiation could significantly reduce the adult lifespan of *T. absoluta*. The activities of catalase (CAT) were affected by UV-A, UV-B, and UV-C compared to the control group. In addition, the UV-A radiation influenced the activities of peroxidase (POD). The result of our findings showed that UV radiation could be a factor affecting growth and development of *T. absoluta.*

## 1. Introduction

Ultraviolet (UV) radiation is a prominent abiotic stressor for numerous organisms, including insects, and has been documented to induce a diverse range of detrimental effects [1,2]. UV radiation can be classified into three types according to distinct wavelength ranges, namely UV-A (320–400 nm), UV-B (280–320 nm), and UV-C (100–280 nm), each exhibiting unique energy levels and biological activities [3,4]. The damage to biological organisms increases as the wavelength of UV light decreases. However, since UV wavelengths below 290 nm are absorbed by the Earth’s ozone layer, over 90% of the UV-A radiation reaches the Earth’s surface [5]. UV-A can induce oxidative stress, whereas UV-B not only induces oxidative stress but also causes direct DNA damage and more severe photoinhibition [6]. UV-C induces responses in insects and causes DNA damage similar to UV-B, but with more pronounced effects [7]. Reactive oxygen species (ROS) production and accumulation can be enhanced by UV radiation. This augmentation leads to increased antioxidant capacity and oxidative products within cells [8]. Low levels of ROS have no detrimental effects on cellular health. Instead, they contribute to maintaining normal biological functions while playing a crucial role in cell signaling and the activation of host defense genes [9]. However, intense UV radiation not only imposes significant selective pressure on insects but also induces changes in their enzyme activity and modifies their genetic material. These effects ultimately result in genetic differentiation among insect populations. Additionally, UV radiation targets various macromolecules such as DNA, proteins, carbohydrates, and lipids within insects’ bodies [8,10,11].

The increasing proportion of UV radiation has sparked heightened scientific interest in the current and potential impacts on herbivorous insects, particularly regarding their survival, development, and behavior. For instance, exposure of the red flour beetle, *Tribolium castaneum* Herbst (Coleoptera: Tenebrionidae), to UV-C irradiation for durations ranging from 4 to 64 min resulted in a significant reduction in gonad size, accompanied by a decrease in both the number of eggs laid and the hatching rate when compared to the control group [12]. Anttila et al. [13] revealed that exposure to UV-B radiation resulted in a significant prolongation of the development time for the *Epirrita autumnata* Borkhausen (Lepidoptera: Geometridae), while no discernible impact was observed on pupal development. Parajuli et al. [14] also reported that UV-A had a negative impact on all stages of *Diaphorina citri* Kuwayama (Hemiptera: Psyllidae) but UV-B was found to be more destructive. However, when adult psyllids were exposed to UV radiation, there was little effect observed on the eggs or nymphs produced. The activity of protective enzymes in insects can be induced by harmful stress, and the enzymes in the body can quickly respond to adverse environmental stress to ensure that the insects can adapt to the environment [15]. Under the induction of toxic secondary substances produced under UV radiation stress, the detoxification and protective enzyme systems in insect bodies are activated, leading to changes in their activities [11]. These changes facilitate the removal or decomposition of toxic substances, thereby protecting insects from potential harm. The influence of UV radiation on antioxidant enzymes and detoxification enzymes was demonstrated by Cui et al.’s findings [16], which revealed irreversible reductions in enzyme activities upon exposure to UV-A, UV-B, and UV-C radiation. Khan et al. [17] found an increase in the activity of superoxide dismutase (SOD) and catalase (CAT) in *Plutella xylostella* Linnaeus (Lepidoptera: Plutellidae), while a decrease was observed in the activity of peroxidase (POD) and carboxylesterases (CarE) with prolonged exposure time. UV radiation has been shown to have the potential to generate high levels of the activities of SOD, POD, and CAT in both *Locusta migratoria tibetensis* Chen (Orthoptera: Oedipodidae) and *Helicoverpa armigera* Hübner (Lepidoptera: Noctuidae) [18,19]. Another study showed that the insect’s enzymatic oxidation system comprises protective enzyme systems, including SOD, POD, and CAT, and this intricate system efficiently eliminates ROS generation, which causes the body and mitigated oxidative damage to insects [20].

*Tuta absoluta* Meyrick (Lepidoptera: Gelechiidae) is an invasive and destructive pest, posing a significant threat to solanaceous crops such as tomatoes, brinjals, and potatoes [21]. The larvae of *T. absoluta* infested all plant organs including leaves, stems, and fruits, resulting in severe damage and yield losses ranging from 80% to 100% in tomato crops [22]. Various strategies have been employed for the management of *T. absoluta*, including mass trapping with sex pheromones, biological control through the application of microorganisms and natural enemies, adoption of agronomic practices, and the use of chemical control measures [23]. Currently, chemical insecticides serve as the primary approach to population management of *T. absoluta*, and the extensive application of these insecticides exacerbates issues related to insecticide resistance [24,25]. Despite the utilization of UV light as an integrated pest management technology for controlling Lepidoptera pests [8]. Does ultraviolet radiation influence the behavior and development of *T. absoluta*? There is currently insufficient experimental evidence to evaluate the impact of ultraviolet radiation on the behavior and development of *T. absoluta*. The objective of this study was to investigate the impact of UV-A, UV-B, and UV-C radiation exposure on the lifespan, reproductive capacity, and antioxidant enzyme activity of *T. absoluta*. The presented data collectively establish a fundamental framework for comprehending the influence of UV radiation on *T. absoluta* by elucidating the underlying molecular mechanisms governing antioxidant enzyme activities.

## 2. Materials and Methods

### 2.1. Insect Rearing

The *T. absoluta* individuals were collected from tomato plants in Hotan City, Xinjiang Province, China. The identification of *T. absoluta* individuals was conducted by extracting DNA, amplifying the samples using CO1 primers, and blasting the sequenced fragments at the NCBI database [26]. From 2021 to 2023, the samples were maintained on tomato plants without exposure to insecticides under controlled conditions of 25 ± 2 °C and 50 ± 10% relative humidity with a photoperiod of 16 h light and 8 h darkness. To conduct the experiment, fifteen pairs of adult *T. absoluta* were individually placed in Petri dishes (*Φ* = 90 mm) to lay eggs. The larvae hatched from eggs and were subsequently reared on tomato leaves until adulthood for further testing. We distinguished between males and females by observing the ovipositor in females, which featured typical mechanoreceptor hairs and could freely extend and retract under a microscope. In contrast, such structures were not observed in males. One-day-old adults were selected and placed in individual petri dishes (*Φ* = 90 mm) with a 10% honey solution. After 24 h, the UV radiation exposure test was conducted.

### 2.2. The Effect of UV Radiation on Biological Parameters of T. absoluta

Ten paired one-day-old adults were randomly selected and exposed to a specific wavelength (UV-A, UV-B, and UV-C) using a lamp (Philips Investment Co., Ltd., Shanghai, China) of radiation for one time as a replicate, and each type of radiation was treated for different durations: 30, 60, 120, and 180 min. The control was not exposed to radiation. The total number of treatments in the experiment were 13, and three replicates were performed for each treatment. After exposure to radiation, a single mating was conducted between adult male and female individuals. The paired insects were placed in a climate chamber with the sustained supply of fresh 10% honey water for eggs laying. The number of eggs laid by an individual female during oviposition and survival individuals was recorded daily until death. The number of eggs was counted under a stereomicroscope.

### 2.3. The Effect of UV Radiation on Antioxidant Enzymes of T. absoluta

The one-day-old adults were randomly selected without differentiating between genders and exposed to a specific wavelength (UV-A, UV-B, and UV-C) of radiation for one time as a replicate. Each type of radiation was treated for different durations: 30, 60, 120, and 180 min. The control was not exposed to radiation. After radiation exposure, the samples were homogenized in a solution at a 1:10 ratio. The procedure of homogenization was performed on ice. The homogenate was centrifuged at 10,000× *g* for 15 min at 4 °C, and the resulting supernatant was subsequently utilized for further analysis. Enzyme activity was immediately quantified using appropriate assays. Each treatment included 20 insects with three biological replicates. The activity levels of superoxide dismutase (SOD), catalase (CAT), and peroxidase (POD) were determined spectrophotometrically by using commercially available assay kits (Suzhou Grace Biotechnology Co., Ltd., Suzhou, China) and following the manufacturer’s protocols. The CAT activity was determined by measuring the decrease in H_2_O_2_, resulting from hydrogen peroxide decomposition, using a UV spectrophotometer to measure the absorbance at 510 nm. One unit of CAT activity was defined as the amount that decomposes 1 μmol of H_2_O_2_ per min per g tissue (U/g). The POD activity was quantified by measuring the absorbance at 470 nm using a UV spectrophotometer, through catalyzing the oxidation of a substrate in the presence of H_2_O_2_. One unit of POD activity was defined as a change in absorbance at 470 nm of 1 per minute in 1 g tissue (U/g). The SOD activity was measured at 450 nm in a UV spectrophotometer by xanthine and xanthine oxidase system. One unit of SOD activity was defined as the amount of enzyme corresponding to an inhibition rate of 50% in the reaction system (U/g).

### 2.4. Statistical Analysis

The experiment employed UV type and radiation duration as independent variables, and biological parameters and antioxidant enzyme activities served as the dependent variables. The relationships between UV type and treatment duration were analyzed with a two-way analysis of variance (ANOVA) and followed by Tukey’s HSD (Honestly Significant Difference) post hoc test (*p* < 0.05) [14]. When interactions were significant, the impact of individual explanatory variables on the response variables was measured separately. If there was a significant difference, Tukey’s method was used for the post hoc test. All processes were completed in SPSS 22.0.

## 3. Results

### 3.1. The Impact of UV Radiation on Biological Parameters of T. absoluta

No significant effect of treatment duration and different UV radiation interactions on lifespan (*F* = 1.44, *df* = 3, *p* > 0.05) and fecundity (*F* = 1.99, *df* = 3, *p* > 0.05) of *T. absoluta* was observed. The adult lifespan of *T. absoluta* was significantly reduced by both UV-A and UV-C radiation at different exposure durations (30, 60,120, and 180 min), with the lowest lifespan of 9.09 d (*F* = 68.30, *df* = 3, *p* < 0.05) observed after 180 min of UV-C irradiation (Figure 1).

There was no significant effect of UV-A and UV-B radiation on egg laying of *T. absoluta*. Exposure to UV-C radiation for 30 and 60 min also did not show any effect, whereas exposure for 120 and 180 min significantly decreased the egg laying of *T. absoluta* by 24.00 eggs (*F* = 8.83, *df* = 3, *p* < 0.05) and 51.20 eggs (*F* = 8.81, *df* = 3, *p* < 0.05), respectively (Figure 2).

### 3.2. The Effect of UV Radiation on the Antioxidant Enzyme of T. absoluta

The UV type and radiation time had no interaction effect on the CAT activity (*F* = 0.92, *df* = 9, *p* > 0.05), but the CAT activity was significant affected by UV radiation type (*F* = 20.70, *p* < 0.05). The CAT activity was observed to decrease in all treatments compared to the control group following exposure to UV radiation (Figure 3). The CAT activity exhibited significant decreases of 30.23% and 38.37% after exposure to UV-B radiation for 30 min and 120 min, respectively (*F*_30 min_ = 7.21, *df* = 3, *p* < 0.05; *F*_120 min_ = 6.95, *df* = 3, *p* < 0.05). Similarly, the CAT activity showed reductions of 17.30% and 30.20% following UV-A radiation for the same time durations. When exposed to UV radiation (UV-A, UV-B, and UV-C) for 60 and 180 min, the CAT activity exhibited a significant decrease compared to the control group (*F*_60 min_ = 24.99, *df* = 3, *p* < 0.05; *F*_180 min_ = 4.32, *df* = 3, *p* < 0.05). Specifically, after 180 min of exposure, CAT activity was decreased by 23.52%, 29.57%, and 26.72% due to UV-A, B, and C radiation, respectively.

The POD activities were significantly affected by UV radiation (*F* = 27.16, *df* = 3, *p* < 0.01), duration (*F* = 8.36, *df* = 3, *p* < 0.05), and their interaction (*F* = 8.74, *df* = 9, *p* < 0.05). The POD activity exhibited a decrease following UV-B and UV-C irradiation compared to the control, but there was no significant difference. UV-A irradiation resulted in a gradual increase in POD activity except for 180 min. The POD activity increased by a factor of 10.44 (*F* = 8.82, *df* = 3, *p* < 0.05) and a factor of 21.17 (*F* = 22.11, *df* = 3, *p* < 0.05), following exposure to UV-A irradiation for 60 min and 120 min, respectively (Figure 4).

The SOD activity was significantly affected by UV treatment (*F* = 19.06, *df* = 3, *p* < 0.05) and durations (*F* = 4.71, *df* = 3, *p* < 0.05), and there was a significant interaction between UV and duration (*F* = 5.98, *df* = 9, *p* < 0.05). The SOD activity exhibited a rising and subsequently declining pattern when exposed to UV-A radiation (Figure 5). The exposure to UV-A radiation for 30 min, 60 min, and 180 min resulted in a respective increase in SOD activity by factors of 1.71 (*F* = 9.02, *df* = 3, *p* < 0.05), 4.54 (*F* = 12.02, *df* = 3, *p* < 0.05), and 1.88 (*F* = 36.23, *df* = 3, *p* < 0.05) compared to the control. The SOD activity gradually increased following UV-B and UV-C irradiation compared to the control; however, there was no statistically significant difference.

## 4. Discussion

Increasing evidence suggests that UV radiation is a ubiquitous environmental stressor with profound impacts on diverse living organisms [15,27]. Previous studies have extensively investigated the effects of UV on various arthropods, consistently demonstrating its direct influence on their survival, development, and behavior across different wavelengths of radiation [28]. For instance, the longevity of *Dialeurodes citri* Ashmead (Hemiptera: Aleyrodidae) adults decreased with increasing exposure time to UV-A [29]. Previous research has shown that adult male and female *D. citri* exhibited reduced longevity after UV-A exposure [14]. Tuncbilek et al. [30] reported a negative correlation between the longevity of *Trichogramma euproctidis* Girault (Hymenoptera: Trichogrammatidae) and increasing UV-C dose. Similarly, Ali et al. [8] demonstrated that exposure to UV-A radiation resulted in a reduction in the lifespan of adult female *Mythimna separata* Walker (Lepidotera: Noctuidae) individuals. Consistent with these findings, our analysis revealed that both UV-A and UV-B radiation significantly decreased the lifespan of *T. absoluta* adults compared to the control group; however, we did not observe any specific impact on lifespan from UV-B radiation alone. Additionally, our results showed a significant decrease in egg laying by *T. absoluta* exposed to 120 min and 180 min of UV-C irradiation. These findings are consistent with previous studies conducted by Collins and Kitchingman [31], Tariq et al. [29], and Tungjitwitayakul et al. [12]. Interestingly, some studies have indicated that both UV-A and UV-B can affect insect fecundity [16,32,33]. The variation in sensitivity to different wavelengths of ultraviolet radiation among insect species may account for this discrepancy. Moreover, defensive enzymes and resistance genes likely play a pivotal role in enhancing their capacity to withstand or mitigate stress induced by ultraviolet radiation while facilitating relocation to more favorable environments for survival.

Relevant studies conducted both domestically and internationally have demonstrated an increase in levels of reactive oxygen species (ROS) in insects following short-term UV stress, with insects exhibiting the ability to adapt to this stress by enhancing enzyme activity for ROS clearance [11]. The enzyme groups, superoxide dismutase (SOD), catalase (CAT), and peroxidase (POD), play crucial roles in antioxidant defense mechanisms and work synergistically to counteract oxidative stress induced by elevated levels of ROS within cells [34]. Li et al. [18] suggested that the activities of SOD, POD, and CAT in insects increased with prolonged exposure to UV-A and UV-B radiation. Our findings indicated an initial increase followed by a subsequent decrease in the activities of SOD and POD under UV-A radiation. Similarly, the activities of SOD exhibited a similar trend under UV-B radiation. These findings suggest that antioxidant enzymes serve as an initial defense mechanism in the short term but have limited protective capacity against oxidative stress [35,36].

In this study, we observed a reduction in the activities of SOD, POD, and CAT upon exposure to UV-C radiation compared to the control group, which is consistent with previous research findings [37]. Cui et al. [16] reported a significant decrease in CAT activity in *Bactrocera dorsalis* Hendel (Diptera: Trypetidae) due to UV-A, UV-B, and UV-C radiation, supporting our analysis. These results also suggest that the antioxidant system may be insufficient to effectively counteract excessive reactive oxygen species production under more severe stress conditions. Moreover, unlike UV-B and UV-C radiation, UV-A radiation did not significantly affect POD and SOD activities. Our findings further validate the harmful effects of both UV-A and UV-B on insect physiology [14,38,39].

ROS synthesis and regeneration were catalyzed by antioxidant enzyme defense mechanisms, which play a pivotal role in insect biology [40]. Previous studies have demonstrated that the enzymatic activities of SOD, POD, CAT, and GST were reduced, resulting in a significant increase in mortality rate among *Spodoptera litura* Fabricius (Lepidoptera: Noctuidae) larvae [41]. Furthermore, exposure to UV radiation significantly increased *B. dorsalis* cohort mortality and prolonged the pre-oviposition period [16]. Our findings indicate that UV-C radiation exposure led to diminished levels of antioxidant enzymes, leading to a substantial decline in both lifespan and egg-laying capacity of *T. absoluta*. However, UV-A radiation exposure notably enhanced antioxidant enzyme activity while concurrently reducing adult lifespan. Insects possess the ability to safeguard their bodies through DNA repair mechanisms against severe UV-radiation-induced DNA damage [42]. This phenomenon may be attributed to specific molecules involved in immune defense that potentially aid in mitigating the impact of UV radiation. Consequently, transcriptomic and metabolomic analyses were conducted to unveil adaptive molecular mechanisms in *T. absoluta* under UV radiation exposure. Our study provides valuable insights into the interactions between *T. absoluta* and UV radiation that could facilitate novel non-polluting approaches to pest species control.

## 5. Conclusions

This study demonstrated that both UV-A and UV-C radiation significantly reduced the adult lifespan of *T. absoluta*. Furthermore, exposure to UV-C radiation for more than 120 min resulted in a substantial decrease in the egg-laying capacity of *T. absoluta*. The activities of CAT were affected by UV-A, UV-B, and UV-C compared to the control group. In addition, the activities of POD and SOD were impacted by UV-A. The study provides crucial insights into the efficacy of UV radiation against *T. absoluta*, and with further research, UV radiation could potentially serve as an effective pest management strategy.

## Figures and Tables

**Figure 1 insects-16-00109-f001:**
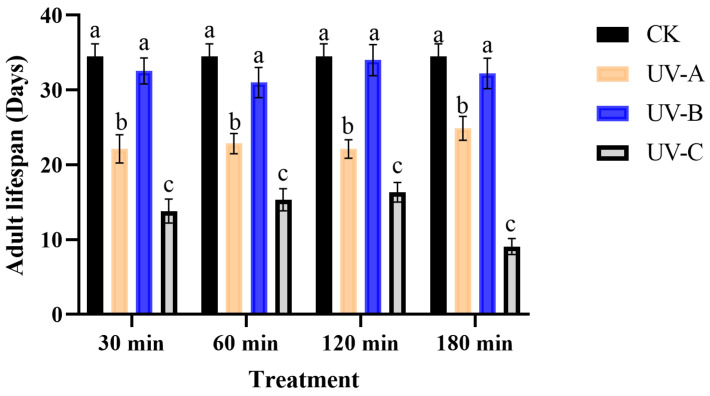
The impact of UV radiation on the longevity of *T. absoluta*. Data are presented as mean ± SE. Different bar labels indicate significant differences (*p* < 0.05) at the same time among different UV radiation according to Tukey’s test.

**Figure 2 insects-16-00109-f002:**
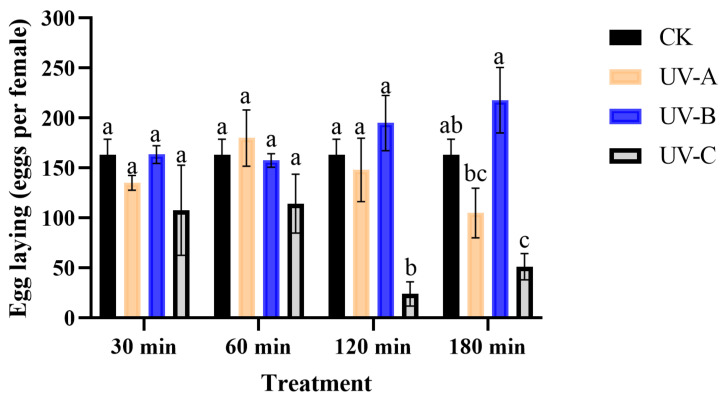
The impact of UV radiation on the egg laying of *T. absoluta*. Data are presented as mean ± SE. Different bar labels indicate significant differences (*p* < 0.05) at the same time among different UV radiation according to Tukey’s test.

**Figure 3 insects-16-00109-f003:**
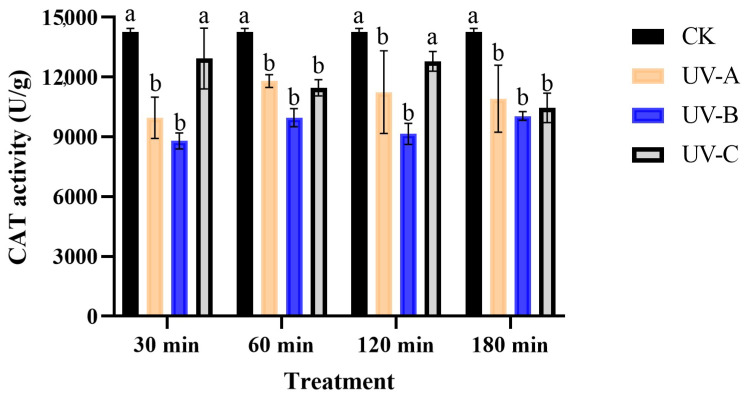
The effect of UV radiation on the CAT activity of *T. absoluta*. Data are presented as mean ± SE. Different bar labels indicate significant differences (*p* < 0.05) at the same time among different UV radiation according to Tukey’s test.

**Figure 4 insects-16-00109-f004:**
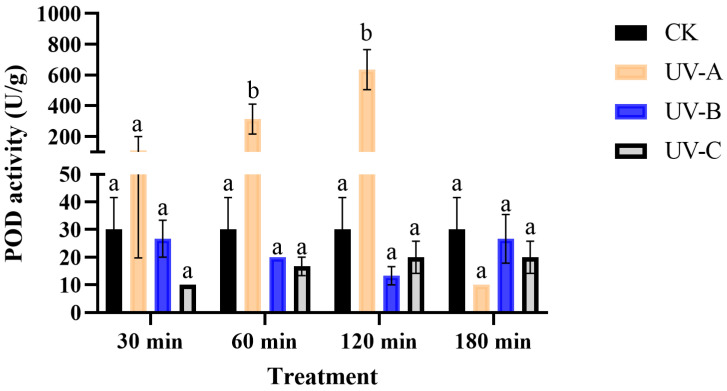
The effect of UV radiation on the POD activity of *T. absoluta*. Data are presented as mean ± SE. Different bar labels indicate significant differences (*p* < 0.05) at the same time among different UV radiation according to Tukey’s test.

**Figure 5 insects-16-00109-f005:**
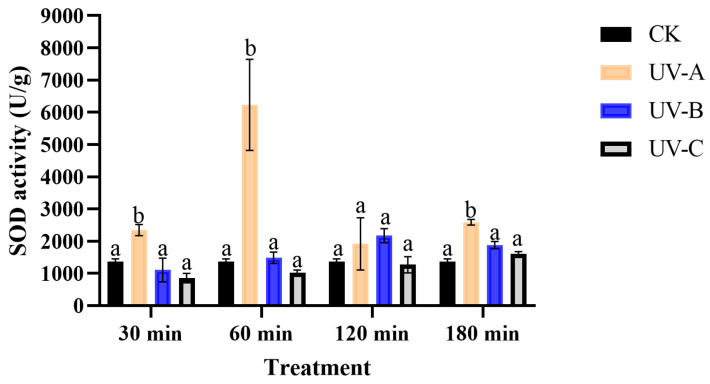
The effect of UV radiation on the SOD activity of *T. absoluta*. Data are presented as mean ± SE. Different bar labels indicate significant differences (*p* < 0.05) at the same time among different UV radiation according to Tukey’s test.

## Data Availability

The data presented in this study are all available in this article.

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
