# Peer review of "Impact of Ultraviolet Radiation on Growth, Development and Antioxidant Enzymes of Tuta absoluta (Meyrick)"

_insects, 2025, doi:10.3390/insects16020109_

Round 1

Reviewer 1 Report

Comments and Suggestions for Authors

These are my main comments on the manuscript (insects-3397885) entitled “Impact of UV radiation on growth, development and antioxidant enzymes of Tuta absoluta (Meyrick)”. This work investigates the effects of three types of UV radiation on the lifespan, egg laying behavior, and antioxidant activities of T. absoluta. Following moderate revisions should be incorporated in the manuscript prior to acceptance. A few points:

L.12: Define UV.

L.13: Tuta absoluta should be in italic.

L.16: … pest infesting of Solanaceae species, was…

L.26: Keywords should be in alphabetic order. Also, keywords serve to widen the opportunity to be retrieved from a database. To put words that already are into title and abstracts makes KW not useful. Please choose terms that are neither in the title nor in abstract.

Ls.51, 53, 59, etc.: For scientific names, provide the ID author, order and family taxa. Correct in all manuscript. 

Ls.75-77: Revise this sentence to eliminate rewordiness.

L.80: Also, a hypothesis for this study is needed.

L.80: Delete “primary”.

Ls.121-128: In this paragraph, sentences should be in past.

Ls.123-124: In particular, the analyses of the biological parameters or antioxidant enzyme activities should include line and treatment, plus the interaction line x treatment as predictors. Explain

Figures 1-5: For ANAVA, comparative results should be classified using letters according to Tukey’s test.

Author Response

Dear reviewer:

We feel great thanks for your comments and professional advice on our article. Based on these comments and suggestions, we have made careful modification on the original manuscript. All changes made to the text are in yellow in the revised manuscript so that you may be easily identified. Some of your questions were answered in appendix material.

Once again, we acknowledge your comments and constructive suggestions very much, which are valuable in improving the quality of our manuscript.

Reviewer 2 Report

Comments and Suggestions for Authors

The conditions under which the control group was kept should be detailed in the Materials and Methods section.

Please add  references to the statistical methods. 

Author Response

We feel great thanks for your comments and professional advice on our article. Based on these comments and suggestions, we have made careful modification on the original manuscript. All changes made to the text are in yellow in the revised manuscript so that you may be easily identified. Some of your questions were answered in appendix material.

Once again, we acknowledge your comments and constructive suggestions very much, which are valuable in improving the quality of our manuscript.

Reviewer 3 Report

Comments and Suggestions for Authors

I appreciate the opportunity to review this interesting article on the relationship between UV radiation and tomato leafminer.

The manuscript titled “Impact of UV radiation on growth, development and antioxidant enzymes of Tuta absoluta (Meyrick)” authored by Zhou et al. provides valuable insights into the interactions between T. absoluta and UV radiation that could be used in integrated pest management.

The methods in the manuscript are not clear, and not well presented. Methods need more description to make the manuscript’s results reproducible based on the details in the methods section.

The manuscript is scientifically sound, and the experimental design is appropriate to test the hypothesis.

The figures are appropriate but not easy to interpret and understand.

The data were interpreted appropriately and consistently throughout the manuscript.

The conclusions are missed.

The cited references are 38 (10 references within the last 5 years; 8 references between 2016 and 2020; 20 references from 2015 and older).

The results are interpreted appropriately. But they are not significant. The life table is very important. It is an approach that is especially useful in entomology, where developmental stages are discrete and mortality rates may vary widely from one life stage to another. What about the eggs? It is not enough to mention the number of eggs. It is necessary to mention the hatching rate, the development of the larvae, their survival, etc.

The question is original and well-defined.

The results provide an advancement of the current knowledge.

The work fits the journal's scope.

The English language is appropriate and understandable.

L2., “UV”, Please use “Ultraviolet”, not UV.

L12. “UV”, Please use “Ultraviolet (UV)”, not UV.

L13., “Tuta absoluta”. It should be in italics.

L26., “Keywords”, Avoid using words from the title.

L90. Please explain the process you use to identify T. absoluta individuals during collections? Add also the used key classification (reference). Including these information will assist guarantee that the process is replicable and well-documented.

L95. Please provide an in-depth description of the rearing of the pairs you collected in order to get adults as a day-old. Were they just reared in the lab for one generation? Please explain how you distinguished between males and females. Please add the reference used as a guide.

L95. “were selected”, On what basis was it chosen? Randomly or what?

L95. “separate cages”, Please add specifications for cages (Size, Material, etc.).

L97., L104., “each radiation is treated for 0”, It cannot be said that it was exposed to radiation for zero seconds. The zero should be removed, and it should be stated that the control was not exposed to radiation.

L97., L104, “each radiation is treated for 0, 30, 60, 120 and 180 minutes, respectively”, It's not clear, I understand that each insect was exposed to this wavelength of radiation for this total amount of time., i.e. 30+60+90+120+180=360 min. Please clarify.

L97. “Three replicates", Please describe what the replicate is. Is it a insect, a cage, or what?

L106., “20 insects”, Are they males or females? Where were they collected from? From insects hatched in the lab or from the field?

L144., “24 eggs”, “51.2 eggs”, Please ensure that the same number of decimal places is used consistently throughout the manuscript.

L150., “There was no significant interaction between UV and duration” This sentence is not clear. Please rephrase it.

L157-158., “When exposed to UV radiation for 60 and 180 minutes, the CAT activity exhibited a significant decrease compared to the control group”, What about 30 minutes?

“Figures 1, 2, 3, 4, and 5”, The marks (*) placed above the columns of the figures to indicate significant differences are not appropriate. They cause confusion and make it difficult to follow the figure. Please change the style, for example, to letters placed above each column. The grayscale of the columns is too close together, making it difficult to distinguish between treatments. Please select another style.

“Figures 2, 3, 4, and 5”, I see under the stars two types of line. For example, in Figure 3, (above the columns of treatment 180 min. there is a straight line under the star) and (above the columns of treatment 180 min. there is a straight line with downward extensions under the star). What is the difference between the two types of lines under the star? Please explain.

“Figures 4, and 5”, Cropping columns makes it difficult to take a comprehensive view and compare treatments with each other. Please add the shapes without cropping.

“Conclusions”, This section is missed. Please add it.

“References”, A crucial error in the references' numbering prevented me from following them. in the list of references, references 1 and 2 were mentioned twice. i am not aware of whether reference number three is three or five. this is also for every reference.

Author Response

(The authors gave the same response as above.)

Round 2

Reviewer 1 Report

Comments and Suggestions for Authors

The authors have incorporated all suggestions and reviewer comments into the latest version, now the manuscript seems much clear. There are minor points to be corrected:

Ls.24-25: Delete “.” after investigation.

L.42:…DNA, proteins, carbohydrates, and lipids…

L.132: Define “HSD”.

Ls.140, 143, 152, 159, 160, 161, 164, 168, 175, 176, 179, 180, 186, 187, 188, 191, 192: Provide the degree freedom.

L.168:…group (F60 min = 24.99, p < 0.05; F180 min= 4.32, p < 0.05)…

L.170:…and C radiation, respectively.

Author Response

Dear reviewer:

We gratefully appreciate for your valuable comments, which has significantly raised the quality of the manuscript and has enable us to improve the manuscript. Each suggested revision and comment, brought forward by the reviewers was accurately incorporated and considered. Below the comments of the reviewers are response point by point.

Once again, we acknowledge your comments and constructive suggestions very much, which are valuable in improving the quality of our manuscript.

Reviewer 3 Report

Comments and Suggestions for Authors

The authors have clearly improved the manuscript and made it really clearer. It is suitable for publication, with minor revisions.

Q5:L90. We sincerely appreciate the valuable comments. The identify of T. absoluta individuals refered to the article titled “First report of the South American tomato leafminer, Tuta absoluta (Meyrick), in China” and “Warning of the dispersal of a newly invaded alien species tomato leaf miner Tuta absoluta (Meyrick), in China (in Chinese)”. We extracted the DNA, amplified the samples using CO1 primers, and blasted the sequenced sequences at NCBI.

Please add this info to the article with the reference that you used.

Q6:L95. T. absoluta were reared in lab from 2021 to 2024. To conduct the experiment, T. absoluta eggs were collected, hatched into larvae, and subsequently reared on tomato leaves until adulthood for testing purposes. We distinguished between males and females by observing the ovipositor in females, which featured typical mechanoreceptor hairs and could freely extend and retract under a microscope. In contrast, males did not observed.

Please add this info to the article.

Q7:L95. We randomly selected it.

Please add this info to the article.

Q11:L97. Ten paired insects were exposed to a specific wavelength of radiation for one time as a replicate.

Please add this info to the article.

Q12:L106. We randomly selected adults

without differentiating between genders. The primary aim of this study was to investigate the effects of UV radiation on defense enzyme activity in adults, which is why we did not distinguish between sexes. They collected from insects hatched in the lab.

Please add this info to the article.

Author Response

(The authors gave the same response as above.)
